# OpenReview forum: "Geometric-Averaged Preference Optimization for Soft Preference Labels"
_NeurIPS.cc/2024/Conference — NeurIPS 2024 poster_

### Official Review · Reviewer_98AM · 2024-06-26

**Soundness:** 3
**Presentation:** 3
**Contribution:** 3
**Rating:** 6
**Confidence:** 4

**Summary:**

Pretrained large language models know information about a wide range of topics, but since pre-training is often done on internet scale data, these models are not aligned with human values. Offline preference learning methods such as DPO are getting increasingly popular for this task.

A key assumption for DPO is that it sees a binary preference dataset  of $(x, y_w, y_l)$ tuples, where $x$ is the prompt, $y_w$ is the preferred and $y_l$ is the dispreferred response. However, this binary labeling is often too hard and does not count for the difference between two responses: while responses $y_1$ and $y_2$ may have a very clear preferred-dispreferred relationship, another pair $y_3$ and $y_4$ might have a much smaller difference.

This paper proposes an algorithm to take into account the relative preference between two responses. The relative preference is not binary and can take any value between 0.5 and 1.0: closer to 1.0 means the preference is almost binary, whereas closer to 0.5 means it is a tossup between two responses. Prior work such as cDPO has used a linear interpolation between two different DPO losses with reversed preference relationships. In comparison, this paper assumes geometric averaging between two LLM policies, and this results in a simple modification to the existing DPO algorithm and its derivatives. This paper’s method, GDPO, outperforms DPO in the case of different soft preference labels.

**Strengths:**

1. The algorithm presented comes from a simple assumption. The final form of the algorithm’s loss functions is simple and intuitive.
2. The paper is nicely written.

In short, this paper comes from a long line of recent papers that try to fix one or more of DPO’s algorithmic issues. The authors focus on the strict binary nature of the preference dataset utilized by DPO, and try to improve it when additional information, i.e., how strong the preference relationship is, is known. This is an interesting question and nicely answered.

**Weaknesses:**

I will list the weaknesses from most important to less important, in my opinion.

**(What about DPO pushing down probabilities of both positive and negative examples)**

The biggest problem with DPO in my opinion, is that it pushes down the log probabilities of both positives and negatives, though it pushes down that of negatives much more, increasing the reward margin. Recent work such as [1] discusses this issue. This makes DPO extrapolate to an OOD region, which might be either good or bad. Discussing the results of [1] in context of this paper’s algorithm would be necessary: we would not want an algorithm that would push down probabilities of winning responses. On the other hand, GDPO’s gradient weighting term, $w_{\theta}$, can actually improve this situation too, and it would be good to know.

Prior work such as RPO [2] has attempted to fix this issue by adding a SFT loss to the DPO loss, whereas SimPO [3] shows that a lot of the mismatches happen because of the reference model’s wrong reward attribution, and removing the reference model from the loss calculation can help. While comparing all of these might be out-of-scope for this paper, at least documenting the basic results from [1], i.e., what happens to the log probabilities of winning vs losing responses under GDPO, DPO and cDPO, would strengthen the paper. Also how does this vary depending on the soft preference distribution?

**(Online DPO)**

Multiple works have shown since DPO that offline preference tuning methods are sub-optimal, and the DPO objective, coupled with samples generated from the model itself, is generally better [1, 4, 5, 6, 7]. These methods can either use the reward from the language model itself to train it [6], or a separate reward model [1]. Could the authors expand this paper’s method to the online variant of DPO? Two problems I can see from the start: in order to use a self-rewarding scheme, the probability/reward from the policy needs to somehow inform us of the soft preference label. Would model miscalibration hurt this? Also for a separate reward model, if not sufficiently strong enough, obtaining the soft preference label can be hard. Also obtaining it during training, for every training sample, can become computationally expensive.


**(More architectures/models tried out)**

Trying this algorithm with more recent models, such as LLama3, or other more commonly used models such as Pythia or Mistral would strengthen the paper. **Note that I do not consider this a major weakness, just something that would make the paper more comprehensive.**

**Questions:**

**Possibility of extension to token level labels**

Could this paper’s method be extended to token level labels? Eg, assume a math problem that has 5 steps. Assume responses $y_1$ and $y_2$ that are both wrong, but wrong in different steps, i.e., $y_1$ gets it correct up till 3rd step, whereas $y_2$ gets it correct till 2nd step. Can the soft preference labels reflect that/can we still make the model learn something here?

# References

[1] Preference Fine-Tuning of LLMs Should Leverage Suboptimal, On-Policy Data, https://arxiv.org/abs/2404.14367

[2] Iterative Reasoning Preference Optimization, https://arxiv.org/abs/2404.19733

[3] SimPO: Simple Preference Optimization with a Reference-Free Reward, https://arxiv.org/abs/2405.14734

[4] Is DPO Superior to PPO for LLM Alignment? A Comprehensive Study, https://arxiv.org/abs/2404.10719

[5] Iterative Preference Learning from Human Feedback: Bridging Theory and Practice for RLHF under KL-Constraint, https://arxiv.org/abs/2312.11456

[6] Self-Rewarding Language Models, https://arxiv.org/abs/2401.10020

[7] Direct Language Model Alignment from Online AI Feedback, https://arxiv.org/abs/2402.04792

**Limitations:**

Yes

---

> ### Author Rebuttal · Authors · 2024-08-07
>
> We thank the reviewer for careful reading and thoughtful feedback.
>
> **> What about DPO pushing down probabilities of both positive and negative examples**
>
> As suggested by the reviewer and following https://arxiv.org/abs/2404.14367, **Figure 1 (f)** in additional PDF measures the log ratio $\log \frac{\pi_{\theta}}{\pi_{\text{ref}}}$ of winner/loser responses and estimated reward gap $\log \frac{\pi_{\theta}(x,y_w)\pi_{\text{ref}}(x,y_l)}{\pi_{\text{ref}}(x,y_w)\pi_{\theta}(x,y_l)}$ on Plasma Plan and Anthropic Harmless. DPO aggressively pushes down the log ratio and increases the reward gap, since DPO objective forces the model to achieve $r_{\theta}(x,y_w)-r_{\theta}(x,y_l) \rightarrow \infty$, which is causing an over-optimization issue. cDPO is more conservative in pushing down the log ratio while leading to worse alignment quality due to objective mismatch. GDPO avoids the issues of such objective mismatch and over-optimization by suppressing the reward gap increase modestly. The log ratio and estimated reward gap with Anthropic Harmless dataset, which has difference soft preference label distribution from Plasma Plan (see Figure 2 in the main text), also show the same trends as seen with Plasma Plan.
>
> **> Online DPO**
>
> As proposed by the reviewer, we conduct the comparison among GDPO, DPO, and cDPO in online (on-policy) settings with the Plasma Plan dataset.
> We prepare 2 variants: (1) incorporating an extra reward model $r_{\psi}(x,y)$ and (2) leveraging estimated self-preference $\rho_{\theta} = \sigma( \text{SG}[\beta\log \frac{\pi_{\theta}(x,y_w)\pi_{\text{ref}}(x,y_l)}{\pi_{\text{ref}}(x,y_w)\pi_{\theta}(x,y_l)}])$, where $\text{SG}$ stands for stop gradient operation. For the extra reward model, we use PaLM 2-XS, the same as a policy LLM.
> **Figure 1 (h)** in the rebuttal PDF provides the results of online alignment methods, which shows that GDPO performs the best in both settings.
> This is because GDPO can cancel the effects from competitive/confusing preference around lower soft preferences such as $\hat{p}=0.5$, which could often help the case when (i) the sampled responses are equally good or (ii) the estimation of preferences are not calibrated enough.
> Especially, GDPO demonstrates significant gain in (2) self-preference settings; in contrast, because the binarization of the preference increases the gap from the true preference, DPO degrades the performance worse.
>
> Please note that, because we adopt PaLM 2-XS as external reward models but offline soft preferences are provided from PaLM 2-L (the number of model parameters are quite different), the online performances do not reach the offline performance. Moreover, we train DPO/GDPO/cDPO in a pure on-policy setting (without any reuse of generated data) and sample only 2 responses per prompt. It would be an interesting future direction to optimize the number of gradient steps to reuse the generated samples (something like “batched iteration” methods) and the number of responses sampled per prompt (e.g. Best-of-$N$ strategy). We will include these results in the revision.
>
> **> More architectures/models tried out**
>
> We provide the additional results with Gemma-2B model (https://arxiv.org/abs/2403.08295), which is an open LLM model with an architecture and pre-training different from PaLM 2-XS; an LLM we mainly used in this paper. **Figure 1 (i)** in additional PDF shows the winning rate on Plasma Plan using Gemma-2B as a base language model. The results show that geometric averaging (GDPO) still outperforms DPO and cDPO on Plasma Plan, Plasma Plan Skewed, and Plasma Plan Stairs datasets. These trends are consistent with those of PaLM 2-XS. We will include these results in the revision.
>
> **> Possibility of extension to token level labels**
>
> As long as the problems token-level DPO can be formulated, we think that GDPO can be extended to such settings. However, token-level formulation might have some technical challenges. For instance, handling different token lengths between a part of $y_1$ and a part of $y_2$ with token-level binary/soft labels might be an issue for both DPO and GDPO. It is another problem that obtaining such a dense token-leven preference signal is more costly than the current settings.

---

> > ### Comment · Reviewer_98AM · 2024-08-07
> >
> > Thanks to the authors for providing a nice rebuttal!
> >
> > A few followup questions:
> >
> > > Please note that, because we adopt PaLM 2-XS as external reward models but offline soft preferences are provided from PaLM 2-L (the number of model parameters are quite different...
> >
> > What are the parameter counts of these two models? What is their agreement rate?
> >
> > > More architectures/models tried out
> >
> > This is a bit unfortunate, as it would be interesting to see if the same performances scale to at least 7 or 8B parameter models, as using them has become quite common for small-scale evaluations. However, if the authors cannot produce these results due to compute constraints, that is understandable as well.
> >
> > > What about DPO pushing down probabilities of both positive and negative examples
> >
> > It seems the GDPO still pushes down probabilities of positives, albeit less compared to regular DPO. It is not clear to me, from the results, how the soft preference label/variation within it actually affects the results. Also it seems it does not mitigate the problem that RPO [1] does.
> >
> > > the online performances do not reach the offline performance.
> >
> > Based on both [2] and [3], online DPO > offline DPO. Are the authors claiming that this is not true (which would be an important finding)? What are the differences in setups that leads to this finding? Or are the authors claiming online DPO > offline DPO, but offline GDPO > online GDPO, possibly because of using a significantly smaller reward model? Clarifying this point would be important.
> >
> > # Questions
> >
> > Any possible extension to a reference model free version, similar to [4]? Conceptual explanation might suffice here. This is not an important question related to the paper, just for my personal curiosity.
> >
> > # References
> >
> > [1] Iterative Reasoning Preference Optimization, https://arxiv.org/abs/2404.19733
> >
> > [2] Is DPO Superior to PPO for LLM Alignment? A Comprehensive Study, https://arxiv.org/abs/2404.10719
> >
> > [3] Preference Fine-Tuning of LLMs Should Leverage Suboptimal, On-Policy Data, https://arxiv.org/abs/2404.14367
> >
> > [4] SimPO: Simple Preference Optimization with a Reference-Free Reward, https://arxiv.org/abs/2405.14734

---

> ### Author Response · Authors · 2024-08-09
> **Response to Reviewer 98AM (1/3)**
>
> We appreciate the quick and detailed response. Please let us know if our follow-up responses address your concerns.
>
> **> What are the parameter counts of these two models? What is their agreement rate?**
>
> The number of parameters in PaLM 2 models is not publicly available in the paper (https://arxiv.org/abs/2305.10403). However, the former version of PaLM models opens their parameter counts (https://arxiv.org/abs/2204.02311). The smallest PaLM has 8B parameters, and the largest PaLM has 540B. PaLM 2 might have similar configurations to PaLM.
>
> For the agreement rate between trained reward models (PaLM 2-XS) and AI rater (PaLM 2-L), we can refer to the preference classification accuracy after the convergence: PaLM 2-XS achieves 90.1% train accuracy and 93.8% validation accuracy. For the agreement rate on **out-of-distribution** data, we can also refer to Figure 4 (right) in the main text. While a preference model analytically recovered from the DPO policy (based on PaLM 2-XS) achieves 94.0% accuracy between PaLM 2-L v.s. Human samples (the same distribution to the training data), it drops the accuracy to 61.6% (PaLM 2-L v.s. GPT-4 samples) and 66.3% (PaLM 2-L v.s. GPT-3.5 samples) when the response pairs are far from training data distribution. This implies that trained external reward models may not be so accurate when DPO/cDPO/GDPO outputs out-of-distribution responses.
>
> **> 2B LLMs are not sufficient. Results with 7-8B LLMs?**
>
> Because the similar size LLMs -- Pythia-1.4B/2.8B (https://arxiv.org/abs/2304.01373) -- are often used in RLHF/offline alignment papers (e.g. DPO: https://arxiv.org/abs/2305.18290, Preference Fine-Tuning of LLMs Should Leverage Suboptimal, On-Policy Data: https://arxiv.org/abs/2404.14367) as a default, we still believe that our additional results from Gemma-2B are enough to claim the scalability of geometric-averaging methods to different LLMs.
>
> However, following the reviewer's request, we can also provide the results from Gemma-7B (https://arxiv.org/abs/2403.08295). We are running the experiments now and please let us follow this up in a few days.
>
> **> What about DPO pushing down probabilities of both positive and negative examples**
>
> First, we -- the author and the reviewer -- can agree that **the log probabilities of preferable/less-preferable samples from the dataset (or their gap) can characterize the training dynamics and algorithm behaviors, but cannot be surrogate metrics proportional to alignment performance**.
>
> Iterative RPO (https://arxiv.org/abs/2404.19733) paper and our experiments provide the evidence to support this:
>
> - (1) Figure 2 (a) in the Iterative RPO paper has shown that SFT achieves a larger log probability of winner samples than RPO (DPO+SFT) with monotonic improvement (i.e. SFT > RPO). However, in terms of performance (test accuracy of GSM8K in Table 1), RPO surpasses SFT (i.e. RPO > SFT). This suggests that it is unclear whether a larger log probability of winner samples correlates to the improvement of target metrics or not.
>
> - (2) Our results (Figure 1 (f) in the rebuttal PDF) also reveal that while the order of log probability of winner samples is cDPO > GDPO > DPO,  the order of the winning rate (Table 1 in the main text) is GDPO > DPO > cDPO.
>
> - (3) The analysis paper (https://arxiv.org/abs/2404.14367) initiating this log probability discussion has only observed the fact that DPO pushes down both preferable log probability and less-preferable log probability, but has not claimed that the increasing trend of the preferable log probability improves the performance.
>
> Moreover, we would like to point out that **it depends on the experimental settings and tasks whether the improvement of preferable log probability happens/is necessary or not**.
>
> - (4) The log probability of winner samples in the preference dataset can achieve its maximum value after optimizing the maximum likelihood objective (i.e. SFT with winner responses). As done in DPO, if the policy is initialized with the SFT checkpoint with preferable responses, further improvement hardly happens because the initial checkpoint is already an almost optimal parameter in terms of winner response likelihood. As explained in Section 4 (L188) in the main text, our experiments have followed the procedure in the DPO paper, starting from SFT checkpoints with preferable responses.
>
> - (5) Iterative RPO has started the experiments from LLaMA-2-70B-Chat checkpoint (Section 3 in the Iterative RPO paper). This is not finetuned with the winner's responses in the GSM8K dataset yet, which means that the policy has a sufficient margin to improve the preferable log-likelihood in the dataset. The maximum likelihood term in iterative RPO methods contributes to increasing the preferable log likelihood.
>
> **(continues to the next thread)**

---

> ### Author Response · Authors · 2024-08-09
> **Response to Reviewer 98AM (2/3)**
>
> **(Continuing from the last thread)**
>
> - (6) The target tasks or metrics can be related to the necessity of the increasing trends of the preferable log-likelihood. For instance, mathematical reasoning or code generation tasks can measure the performance with exact match/unit tests and need to maintain reasonable responses, because the best responses do not significantly differ from the preferable responses in the dataset. In these tasks, the degradation of preferable outputs in DPO may cause significant issues. DPO v.s. PPO paper (https://arxiv.org/abs/2404.10719) has stated `However, we will demonstrate in Sec. 6 that even with a nearly perfect annotator, the performance of DPO remains unsatisfactory in challenging tasks such as code generation.` (from Section 4.3: Practical Remark). In contrast, our paper has focused on "open-ended generation" tasks such as summarization or conversation, where we cannot measure the performance with an exact match, rather requiring a human or AI rater to evaluate the quality. In these tasks, the exploration into out-of-distribution regions makes more sense, and the policy needs to push down the likelihood of the (both winner and loser) responses to further improve the response quality.
>
> In summary, we believe that our additional experiments (Figure 1 (f) in the rebuttal PDF) characterize the behavior of geometric averaging well;  cDPO suffers from objective mismatch due to the conservative update (both log probabilities do not push down much, and the winning rate is the worst). DPO faces an over-optimization issue induced by the maximization objective $r_{\theta}(x,y_w)-r_{\theta}(x,y_l) \rightarrow \infty$. This is aligned with our experimental observations (both log probability pushes down the most and the largest reward gap. The performance is the second.) GDPO resolves those two issues by adjusting the scaling factor of the gradient with soft labels (the decrease of log probability suppresses, and the reward gap stays in a modest range. The performance is the best). The trend of the log probability/reward gap is not proportional to the alignment quality. They need to be in a reasonable range. Because the policy has been initialized with SFT models finetuned with (50%) winner responses in the dataset, there is no margin for any algorithms to increase the preferable log-likelihood further. RPO increases log probability, but this comes from the experimental settings. The RPO paper has started the experiments from the checkpoints, which are not finetuned with the target tasks and have enough margin for improvement.
>
> Lastly, we provide the table to compare the relationship among log-likelihood, reward gap, and the binary winning rate (the results are the same as Figure 1 (f) in the rebuttal PDF).
>
> ||400|800|1200|1600|2000|
> |--|--|--|--|--|--|
> |DPO (Pref ratio)|-3.97|-5.23|-7.77|-9.96|-11.9
> |cDPO (Pref ratio)|-1.14|-1.46|-1.79|-2.00|-2.00
> |GDPO (Pref ratio)|-3.24|-3.38|-4.49|-5.83|-9.18
> |DPO (R gap)|33.7|38.4|43.3|49.0|54.5
> |cDPO (R gap)|13.6|14.7|15.3|16.0|16.1
> |GDPO (R gap)|23.1|26.6|29.4|33.5|40.8
> |DPO (WR)|68.18|73.29|78.75|81.53|83.16
> |cDPO (WR)|51.57|61.41|70.62|75.96|72.13
> |GDPO (WR)|71.66|75.49|80.95|84.55|85.48
>
> We appreciate the reviewer deepening the discussion. We are happy to address your concerns if you have any.
>
>
> **> The online performances do not reach the offline performance.**
>
> We'd like to clarify that our additional experiments (Figure 1 (h) in the rebuttal PDF) intend to verify the scalability of our geometric-averaging methods to online (on-policy) settings; i.e. online GDPO > online DPO or online cDPO, and do not intend to compare/discuss the performance between offline methods and online methods. The results reveal that, as the same as offline settings, online GDPO consistently outperforms online DPO and cDPO in both the extra reward model and self-rewarding settings.
>
> For the discussion about online DPO and offline DPO, we would also like to clarify that their performances significantly depend on the scale of reward models (i.e. the preference label quality). In the Online AI Feedback paper (https://arxiv.org/abs/2402.04792), Figure 3 shows that, when the preference labels are annotated by PaLM 2-L, online DPO achieves a 95% winning rate against the SFT model on Reddit TL;DR dataset, and offline DPO achieves 90% winning rate. In contrast, Figure 5 also shows that even online DPO achieves a lower winning rate when the scale of reward models is small; online DPO with PaLM 2-S achieves an 86% winning rate against the SFT model (blue bars) and online DPO with PaLM 2-XS achieves 82%.
>
> Because our online experiments use PaLM 2-XS for the extra reward model and offline experiments use PaLM 2-L, our performance gap between online and offline methods is consistent with the previous literature. Our experiments emphasize the scalability of geometric averaging under the same conditions.
>
> We are glad to provide further clarification upon request.

---

> ### Author Response · Authors · 2024-08-09
> **Response to Reviewer 98AM (3/3)**
>
> **> Reference Model Free Version (Geometric SimPO)**
>
> We think we can apply geometric averaging to the reference model free SimPO. We start with the DPO objective without a reference model (we omit the input $x$ for readability). The derivation of the objective is as follows:
>
> $E[\beta \log\pi_{\theta}(y_w) - \beta \log\pi_{\theta}(y_l) ]$
>
> $= E[\beta \log \frac{\pi_{\theta}(y_w)^{\hat{p}}\pi_{\theta}(y_l)^{1-\hat{p}}/Z_{w}}{\pi_{\theta}(y_l)^{\hat{p}}\pi_{\theta}(y_w)^{1-\hat{p}}/Z_{l}}]$
>
> $= E[\beta (2\hat{p}-1) \log\pi_{\theta}(y_w) - \beta (2\hat{p}-1) \log\pi_{\theta}(y_l) - \beta\log\frac{Z_{w}}{Z_{l}}]$
>
> where $Z_{w}$ and $Z_{l}$ is a partition function. Because they are hard to estimate accurately, we may treat the term $\beta\log\frac{Z_{w}}{Z_{l}}$ as a constant value $\gamma \geq 0$. By adding length normalization coefficients, we have an objective for Geometric SimPO.
>
> $L_{\text{GSimPO}} := E[\frac{\beta(2\hat{p}-1)}{|y_w|} \log\pi_{\theta}(y_w) - \frac{\beta(2\hat{p}-1)}{|y_l|} \log\pi_{\theta}(y_l) - \gamma]$.

---

> ### Comment · Reviewer_98AM · 2024-08-09
> **Followup from Reviewer 98AM**
>
> I thank the authors for the prompt response!
>
> I agree with the points the authors present. As long as the authors present the log-probability argument in this paper in the final version of the paper, along with discussions of the following papers:
>
> [1] Iterative Reasoning Preference Optimization, https://arxiv.org/abs/2404.19733
>
> [2]  Preference Fine-Tuning of LLMs Should Leverage Suboptimal, On-Policy Data, https://arxiv.org/abs/2404.14367
>
> [3] From $r$ to $Q^*$: Your Language Model is Secretly a Q-Function, https://arxiv.org/abs/2404.12358
>
> I am satisfied with the results. The reason I insist on this is: It is certain now that DPO over-extrapolates to an OOD region. Whether this OOD region is good or bad depends on the task: for math problems, as [1] shows, it is probably bad, because only a narrow distribution of responses is good, whereas for RLHF tasks that this paper considers, this might be okay. **It is important to note the effect of new variations of DPO in this regard, specially their learning dynamics.**
>
> Section 5.3 of [3] seems to show a theoretical understanding of why this happens, and possibly including a discussion of this in the paper should improve the quality of this work.
>
> I am willing to increase my scores once the authors produce results on models of around 7B parameter count range.
>
> Kudos to the authors for leading a nice and thorough rebuttal!

---

> ### Author Response · Authors · 2024-08-12
>
> **> Discussion about Log Ratio and Training Dynamics**
>
> We thank the reviewer for discussing this thoroughly. We also agree with your final recap and will include all the experimental results, discussions, and references you provided during the rebuttal in the revised paper.
>
> **> Winning Rate on Plasma Plan, Skewed and Stairs with Gemma-7B**
>
> In the following table, we compare the winning rate of Gemma-7B among DPO, cDPO, and GDPO (against PaLM 2-L). The results show that, following the results on PaLM 2-XS and Gemma-2B, applying soft labels and weighted geometric averaging improves the alignment quality compared to the binary methods (DPO) and conservative methods with soft labels. Geometric averaging can be effective for various model sizes and architectures.
>
> | Method | Plasma Plan (Binary) | (Percentage) | Skewed (Binary) | (Percentage) | Stairs (Binary) | (Percentage)
> |--|--|--|--|--|--|--|
> |SFT| 42.62% | 44.45% | 42.62% | 44.45% | 42.62% | 44.45% |
> |DPO| 79.56% | 61.53% | 78.63% | 61.47% | 75.49% | 60.12% |
> |cDPO| 74.33%| 59.91% | 73.52% | 56.79% | 71.89%| 59.91% |
> |GDPO| 82.58% | 64.11% | 82.23% | 63.73% | 80.37% | 62.61% |
> |$\Delta(+\text{Geom.})$| +5.64% | +3.39% | +6.16% | +4.60% | +6.68% | +2.60%|
>
> Lastly, thank the reviewer again for your active engagement with the rebuttal and discussion despite the limited time. Your thoughtful feedback helped improve our paper.

---

> > ### Comment · Reviewer_98AM · 2024-08-12
> >
> > Dear Authors,
> >
> > Thanks a lot for the updated results! This is satisfactory to me, I am increasing the score from 5 to 6, since **the authors have updated the rebuttals with results of 7B parameter range LLMs, as discussed before.**
> >
> > Kudos on driving a successful rebuttal.

---

### Official Review · Reviewer_EtLq · 2024-07-10

**Soundness:** 3
**Presentation:** 2
**Contribution:** 3
**Rating:** 5
**Confidence:** 2

**Summary:**

The paper proposes a variation to Direct Preference Optimization which takes into account "soft labels," which aim to reflect that not every evaluator might agree on the relative ordering of a pair of model outputs, or that there may otherwise be a lack of confidence in the ordering of outputs. They model this as the likelihood that one output is better than another and use this likelihood in selecting the winner from two prompts using geometric averaging of the confidence in each output.

This approach is straightforwardly extended to several algorithms in the DPO family. This is then used in fine-tuning a model. Experiments compare their variations of DPO with other DPO algorithms and show that an LLM evaluation agent prefers their model outputs over those of PaLM-2L and GPT4 more often than the evaluator prefers other DPO outputs.

**Strengths:**

The central idea of the paper is well motivated and approaches the problem in what seems a quite reasonable manner. I found the authors generally seemed to do a good job of explaining why they were doing what they did in the design of GDPO.

Sections 2 and 3, the primary technical sections, are fairly understandable and clearly written.

I am not entirely convinced that the experiments are optimal but they do suggest that this method provides significant improvements over existing approaches.

I am not well positioned to evaluate the originality of this work. That said, I am not aware of other existing work that approaches this problem in the same way.

**Weaknesses:**

I am only loosely familiar with the current research in this area but I am aware of criticism of the Bradley-Terry model. While this might help to learn preferences under the BT model, it might aid the paper to include a comment on whether this model is worth continued analysis.

Some aspects of the paper could be written more clearly to explain what is happening or why. In particular, I find the paper likely assumes that readers are extremely familiar the most recent approaches to LLM evaluation and corresponding norms of how to present these results. Not being familiar with these things, I feel there are several aspects of the paper that could use more explanation; some are listed below.


The definition of p^ is vital to a thorough understanding of the paper. Right now, I find it's meaning somewhat difficult to parse when there are no subscripts. The paper would be improved by a more clear explanation of what it means in context. This would also make Figure 2 and some of Figure 1 more clear.

Similarly, in Eq. 10, y_w and y_l seem disconnected from y_1 and y_2. I generally understand what's going on but a more clear explanation would aid many readers.


The explanation of exactly what models you applied your techniques to may be clear to someone actively engaged in very similar research but was unclear to me.

Eq 17 needs clarification. From study it's meaning can be interpreted but seeing as both y's come from llm's (as far as I understand) and are involved in testing it is not immediately clear which meaning to assign to y_llm and y_test.


Some of the minor editing issues:

Sentence on line 113 is not correct.

line 129 - "Bradly-Terry"

line 158/159 - maybe "500,000 training instances are ..."?

**Questions:**

Can you expand on the reasonableness of using an LLM to judge LLM performance? Can this lead to potential damage to model training in the future?

**Limitations:**

See question above.

---

> ### Author Rebuttal · Authors · 2024-08-07
>
> We thank the reviewer for careful reading and detailed feedback.
>
> **> W1 (Problem from Bradley-Terry model)**
>
> As the reviewer mentioned, the objective functions stemming from Bradley-Terry model cause "over-optimization" issues, which is inherent in DPO and its derivation from Bradley-Terry model. Recalling the objective of DPO and reward models, it is $\max \log p_{\theta}(y_1 \succ y_2 | x) = \max \log \sigma(r_{\theta}(y_1) - r_{\theta}(y_2))$ (as explained in Section 2, L58 & L83). Since $p_{\theta}$ is a probability, the maximization objective forces $p_{\theta}(y_1 \succ y_2 | x) \rightarrow 1$.
>
> As discussed in Section 3.2, because geometric averaging cancels the gradient from equally good samples (e.g. p=0.55), our methods can successfully mitigate over-optimization issues. Please see **Figure 1 (f)** in additional PDF; we visualize that GDPO suppresses the reward gap increase modestly. It is an orthogonal but important future direction to derive other better preference modelings instead of the Bradley-Terry model.
>
> **> W2 (Clarity)**
>
> We will follow your suggestions to improve clarity of this paper.
>
> **> W2.1**
>
> First, following another reviewer's feedback, we will update the definition of soft preference labels as follows in the revision:
>
> ---
>
> We assume that the binary preference labels ($l(y_1 \succ y_2 | x) = 1$) are sampled from the Bradley-Terry model preference distribution with the parameter $p^{\star}(y_1 \succ y_2 | x) = \sigma(r^{\star}(x,y_1) - r^{\star}(x,y_2))$.. We define soft preference labels as estimates of the true preference probability: $\hat{p}_{x,y_1,y_2} := \hat{p}(y_1 \succ y_2 | x) \approx p^{\star}(y_1 \succ y_2 | x)$.
> For instance, we can estimate this via monte-carlo sampling such as:
>
> $\hat{p} = \frac{1}{M}\sum_{i=1}^{M} l_i (l_i \in \{0,1\})$,
>
> which is done via majority voting among $M$ people in practice. The sampled binary preference may sometimes flip with probability $\epsilon$ (i.e. label noise). If the label noise is known, we may consider the expectation over the noise such as: $\hat{p} = (1-\frac{1}{M}\sum_{i=1}^{M} \epsilon_i) \frac{1}{M}\sum_{i=1}^{M} l_i  + \frac{1}{M}\sum_{i=1}^{M} \epsilon_i \frac{1}{M}\sum_{i=1}^{M} (1 - l_i)$, or we may ignore the noise if $\epsilon_i$ is small and $M$ is sufficiently large.
> Alternatively, we can also estimate the soft preference directly via Bradley-Terry models with some reward function. The direct estimation is often adopted in AI feedback with scoring.
>
> ---
>
> In this paper, $\hat{p}$ always means $\hat{p}(y_1 > y_2 | x)$ to reduce the redundancy and emphasize $\hat{p}$  is a given label from the dataset. We will clarify this and include subscripts appropriately depending on the context.
>
>
> **> W2.2**
>
> $y_w$ represents the winner response, and $y_l$ represents the loser response. As stated in Section 2, we assume $y_1 \succ y_2$ always holds in this paper unless otherwise mentioned (i.e. $y_w=y_1$, $y_l=y_2$). In Equation 10, to emphasize that the weighted geometric average is taken for the distribution, we employ $y_w$ and $y_l$ instead of $y_1$ and $y_2$. We will clarify this in the revision.
>
>
> **> W2.3**
>
> For the base LLM we used, we have clearly stated in the beginning of Section 4: “In the experiments, we use PaLM 2-XS for the base LLM”. Furthermore, we provide additional results with the open Gemma-2B model. Please see **Figure (i)** in additional PDF for the results.
>
>
> **> W2.4**
>
> Thank you for pointing out the notation. $y_llm$ stands for the response from the models trained with DPO/cDPO/GDPO, etc. $y_test$ stands for the reference response from PaLM 2-L, GPT-4, or humans, which is used for the evaluation. We will fix the notation from $y_{llm}$ to $y_{gen}$ (i.e. generated response), and $y_{test}$ to $y_{ref}$ (i.e. reference response) for clarity.
>
> **> W3**
>
> We also thank you for pointing out the minor editing issues. We will fix them appropriately in the revised paper.
>
>
> **> Q1**
>
> First, because our formulation of soft preference labels and geometric averaging are not limited to AI feedback (e.g. majority voting from humans), we believe that the discussion whether it is reasonable for LLM to judge LLM performance is out of scope from this paper.
> However, we can justify the LLM-as-a-judge with some evidence. **Figure 1 (c)** in rebuttal PDF provides the agreement evaluation between human and LLM judges on Plasma Plan. We compare the responses from PaLM 2-L and GPT-3.5. The agreement accuracy reaches 81.3\%. This observation is also consistent with previous literature (https://arxiv.org/abs/2306.05685, https://arxiv.org/abs/2309.00267). In addition, as mentioned in Section 4.2 (L200), LLM rating has a position bias (https://arxiv.org/abs/2305.17926), and to mitigate this, we take the average of $\hat{p}_{\text{AI}}$ by flipping the ordering of ($y_1$, $y_2$) in the evaluation prompt.
>
> Since this paper focuses on the RLHF/alignment phase, rather than pre-training, we think that the catastrophic degradation by the synthetic data would not happen. We will add these related discussions to Appendix in the revision.

---

> > ### Comment · Reviewer_EtLq · 2024-08-08
> >
> > Thank you for the response. My question was not so much asking "does it currently work to use LLMs as judges" but, rather, what the impact of this can be in the future. It is, of course, expected that papers should consider what effects they may have on the world as a result of being published.

---

> ### Author Response · Authors · 2024-08-09
> **Response to Reviewer EtLq**
>
> We appreciate the quick response and clarification. The potential social impacts have been discussed in Appendix A,  "Broader Impacts" section (in the original submission), and following your clarification, we can further extend it considering the effect of AI feedback or training data synthesized by LLMs in the future.
>
> The use of LLMs for AI feedback and synthetic data generation has significantly reduced the costs associated with manual annotation and data curation, enabling scalable learning. However, it remains unclear whether LLMs can accurately identify biases such as prejudice and discrimination when providing AI feedback; potentially LLMs wrongly provide preference labels leading to undesirable biases. Additionally, while agreement between human and LLM preferences is generally high (around 80-85%), the remaining 20% of disagreements could contribute to the accumulation of errors through iterative feedback processes, amplifying the less preferred preferences. Continuous human monitoring is therefore crucial to ensure safety and mitigate potential risks. Furthermore, learning with synthetic data, particularly in pre-training, has shown potential for catastrophic performance degradation due to data distribution shifts. It is also important to be mindful of potential performance deterioration during post-training phases, including alignment, when using synthetic data.
>
> We will include these discussions in the revised version.

---

### Official Review · Reviewer_msmA · 2024-07-12

**Soundness:** 3
**Presentation:** 3
**Contribution:** 2
**Rating:** 6
**Confidence:** 4

**Summary:**

This paper introduces a novel approach to aligning Large Language Models (LLMs) with human preferences by incorporating distributional soft preference labels. The authors argue that existing methods like Direct Preference Optimization (DPO) assume binary, deterministic preferences, which may not accurately reflect the nuanced nature of human judgments. To address this, they propose a modification to DPO that uses a weighted geometric average of LLM output likelihood in the loss function, allowing for a more nuanced representation of preferences. The method can be easily applied to any DPO-based algorithm and shows consistent improvements on standard alignment benchmarks, particularly for data with modestly-confident labels. The authors simulate soft preference labels using AI feedback from LLMs in their experiments.

**Strengths:**

The approach of adapting different scaling factors in preference optimization is a handy and intuitive method.

**Weaknesses:**

The paper lacks sufficient theoretical analysis to justify the use of "Geometric Averaging" for soft labels in preference optimization. While the authors compare their method to others in terms of scaling factors, they fail to provide an in-depth analysis of its effectiveness. Key questions remain unanswered: What are the core weaknesses of previous scaling factors? Why are scaling factors crucial in preference optimization? How does the proposed method fundamentally differ from prior work? The paper would benefit from a more rigorous examination of the method's impact on training dynamics, such as reward gaps and convergence stability. Without this analysis, it's challenging to fully understand and evaluate the method's contributions to the field.

**Questions:**

The soft label addresses preference variability, which relates to label noise. While the authors discuss robustness to noise, it's unclear how this approach actually performs when faced with inconsistent or noisy preference labels. Can the authors provide a more detailed analysis of the method's robustness to label inconsistencies? This would help clarify the practical advantages of soft labels in real-world scenarios where human preferences may be inconsistent or contradictory.

**Limitations:**

The authors adequately discussed the limitations in the seperated section.

---

> ### Author Rebuttal · Authors · 2024-08-07
>
> We appreciate the thoughtful feedback. Please let us know if our responses in the following address your concerns.
>
> **> Main Weaknesses of Previous Papers & How GDPO Resolves them**
>
> As discussed in Section 3.2,  GDPO can avoid (1) "over-optimization" issues (compared to DPO) and (2) objective mismatch between text generation and preference modeling (compared to cDPO), which are fundamental issues of previous works.
>
> (1) Binary preference methods such as DPO, suffer from "over-optimization" issues, where the maximization objective forces $p_{\theta}(y_1 \succ y_2 | x) \rightarrow 1$ (even if the soft label is around 0.5) and induces $r_{\theta}(y_1) - r_{\theta}(y_2) \rightarrow \infty$. This is also raised in the IPO paper (https://arxiv.org/abs/2310.12036). In contrast, GDPO can ignore the gradient from less-confident samples (as shown in Figure 1 (left)), which can mitigate over-optimization.
>
> (2) Soft preference methods with linear interpolation such as cDPO, suffers from objective mismatch between text generation and preference modeling. We have shown that in the synthetic experiments in Figure 1 (right), and also have demonstrated that the same phenomena happen in LLM experiments (Figure 4 (right)). By maintaining the scale of gradient when soft labels are large and ignoring the update when the paired responses are equally good, GDPO avoids the issue of objective mismatch.
>
> **> Analysis on Training Dynamics**
>
> **Figure 1 (f)** in the additional PDF shows the log ratio $\log \frac{\pi_{\theta}}{\pi_{\text{ref}}}$ of winner/loser responses and estimated reward gap $\log \frac{\pi_{\theta}(x,y_w)\pi_{\text{ref}}(x,y_l)}{\pi_{\text{ref}}(x,y_w)\pi_{\theta}(x,y_l)}$ on Plasma Plan and Anthropic Harmless (please refer to https://arxiv.org/abs/2404.14367 for discussion of these metrics). We can see that DPO aggressively pushes down the log ratio and increases the reward gap, since DPO objective forces the model to achieve $r_{\theta}(x,y_w)-r_{\theta}(x,y_l) \rightarrow \infty$, which is causing an over-optimization issue. cDPO is more conservative in pushing down the log ratio while leading to worse alignment quality due to objective mismatch. GDPO avoids the issues of such objective mismatch and over-optimization by suppressing the reward gap increase modestly.
>
> **> Theoretical Analysis on the Optimality Bound**
>
> We here provide a corollary derived from Theorem 4.1 from https://arxiv.org/abs/2406.01462, which shows the bound of optimality gap is improved by GDPO: from $O(C\sqrt{\epsilon_{dpo}})$ (DPO) to $O(C\sqrt{\epsilon_{dpo} - \epsilon_{\bar{p}}})$ (GDPO). Due to the word limit, we omit some assumptions presented in the previous paper, but we are happy to follow-up in the discussion period upon request. We will also include these descriptions in the revision.
>
> First of all, we make the following two assumptions:
>
> **Assumption 1** (Overestimation of the learned reward):
> For all $x,y^{1},y^{2} \sim \rho \circ \pi_{\text{ref}} \text{ s.t. } y^{1} \succ y^{2}$ and the learned reward function $\hat{r}$, we have
>
> $r^\star(x,y^{1}) - r^\star(x,y^{2}) \leq p^\star(y^{1} \succ y^{2} | x)(\hat{r}(x,y^{1}) - \hat{r}(x,y^{2})).$
>
> **Assumption 2** (Relation between GDPO and DPO):
> For the learned reward from GDPO $\widehat{r_{\text{gdpo}}}$ and DPO $\widehat{r_{\text{dpo}}}$, we assume that
>
> $\Delta \widehat{r_{\text{gdpo}}}=(2p^\star(y^1\succ y^2|x)-1) \Delta \widehat{r_{\text{dpo}}}$.
>
> where $y^1\succ y^2$ and $\Delta\widehat{r} = \widehat{r}(x,y^1)-\widehat{r}(x,y^2)>0$.
>
> **Corollary 1**: Let $\pi_{\text{ref}}$ be any reference policy such that global coverage assumption (from previous paper) holds. For any policy $\pi_{\text{dpo}}$ such that the event in the assumption of in-distribution reward learning (from previous paper), Assumption 1 and 2 holds, we have that
>
> $E_{x\sim\rho} [E_{y^1,y^2 \sim \pi_{\text{ref}}( \cdot | x) \text{ s.t. } y^1 \succ y^2} [( r^\star(x,y^1) - \widehat{r_{\text{gdpo}}}(x,y^1) - r^\star(x,y^2) + \widehat{r_{\text{gdpo}}}(x,y^2)^2] ] \leq \epsilon_{\text{dpo}} - \epsilon_{\bar{p}}$,
>
> then we have
>
> $ J(\pi^\star)-J(\pi_{\text{gdpo}})\leq O(C\sqrt{\epsilon_{\text{dpo}} - \epsilon_{\bar{p}}}).$
>
> Since $ J(\pi^\star)-J(\pi_{\text{dpo}}) \leq O(C\sqrt{\epsilon_{\text{dpo}}})$, the bound is improved by GDPO.
>
> **Proof of Corollary 1**
>
> $E_{x\sim\rho}[E_{y^1,y^2\sim\pi_{\text{ref}}(\cdot|x)}[(r^\star(x,y^1)-\widehat{r_{\text{gdpo}}}(x,y^1)-r^\star(x,y^2)+\widehat{r_{\text{gdpo}}}(x,y^2))^2-(r^\star(x,y^1)- \widehat{r_{\text{dpo}}}(x,y^1)-r^\star(x,y^2)+\widehat{r_{\text{dpo}}}(x,y^2))^2]]$
>
>
> $=E_{x\sim\rho}[E_{y^1,y^2\sim\pi_{\text{ref}}(\cdot|x)}[\Delta\widehat{r_{\text{gdpo}}}^2+2\Delta r^\star(\Delta\widehat{r_{\text{dpo}}}-\Delta\widehat{r_{\text{gdpo}}})-\Delta\widehat{r_{\text{dpo}}}^2]]$
>
> $=E_{x\sim\rho}[E_{y^1,y^2\sim\pi_{\text{ref}}(\cdot|x)}[4(1-p^\star(y^1\succ y^2|x))\Delta\widehat{r_{\text{dpo}}}(\Delta r^\star - p^\star(y^1\succ y^2|x) \Delta\widehat{r_{\text{dpo}}})]]$
>
> $\leq0$,
>
> then some small $\epsilon_{\bar{p}}\geq 0$ exists such that
>
> $E_{x\sim\rho}[E_{y^1,y^2 \sim \pi_{\text{ref}}(\cdot|x) \text{ s.t. } y^1\succ y^2} [(r^\star(x,y^1) - \widehat{r_{\text{gdpo}}}(x,y^1) - r^\star(x,y^2)+\widehat{r_{\text{gdpo}}}(x,y^2)^2] ]+\epsilon_{\bar{p}}\leq\epsilon_{\text{dpo}}.$
>
> **> Performance under Label Noise**
>
> In **Figure 1 (g)** in the additional PDF, we provide the winning rate on Plasma Plan with different label noise $\epsilon$. Please also see the response to Reviewer **GdY5** for the definition of label noise. We assume flipping binary label (B-Flip), soft label (S-Flip) with probability $\epsilon$, and taking the expectation of soft labels with probability $\epsilon$ (S-Ave.). While DPO is often affected, GDPO mitigates the noise and performs the best in all the cases.

---

> > ### Comment · Reviewer_msmA · 2024-08-11
> > **Response to the Authors**
> >
> > Thank you for your response.
> >
> > After reviewing the authors' replies to the other reviewers and following the ongoing discussions, I find the concept of "soft preference labels with AI feedback" quite interesting, and I have raised my score accordingly.
> >
> > I acknowledge that the proposed method effectively addresses some algorithmic challenges in DPO, and the analysis conducted on it appears promising. My main concern now is whether the TLDR and Anthropic HH datasets are sufficient to fully validate the method's efficacy. However, I understand that this limitation is unavoidable given the scarcity of available public data, so I did not consider this part when deciding my score.

---

> ### Author Response · Authors · 2024-08-12
>
> We thank the reviewer for reading the rebuttal and the thoughtful consideration.
>
> For the benchmark, we believe that Reddit TL;DR and Anthropic Helpfulness and Harmlessness datasets are widely adopted in previous RLHF/alignment works (e.g. https://arxiv.org/abs/2305.10425, https://arxiv.org/abs/2309.06657, https://arxiv.org/abs/2309.16240, and more). As the reviewer mentioned, because the soft labels have not been used much ever and the community did not have suitable public dataset, we needed to build our experiments on top of popular configurations. We hope our work inspire the community to leverage the soft labels and release the open dataset with soft label annotations in the future.
>
> We will include all the discussion with the reviewer (analysis on training dynamics & theoretical analysis) in the revision. Thank you again for taking your time.

---

### Official Review · Reviewer_GdY5 · 2024-07-13

**Soundness:** 2
**Presentation:** 2
**Contribution:** 3
**Rating:** 6
**Confidence:** 2

**Summary:**

This paper introduces the concept of soft preference labels and proposes leveraging this distributional information, alongside deterministic binary preferences, to enhance Direct Preference Optimization (DPO) losses. By incorporating a weighted geometric average of the LLM output likelihood in the loss function, the authors demonstrate that geometric averaging consistently improves performance on standard benchmarks.

**Strengths:**

1. The paper proposes a novel method to incorporate additional distributional information that is sometimes available in vote pooling data curation processes (assuming this definition is correct).

2. The demonstration of the scaling factor $w_\theta$ in relation to the gradient provides a more intuitive understanding of why this method improves performance.

3. The paper includes fairly extensive evaluation studies.

**Weaknesses:**

The writing in general is very hard to follow and at times very unclear. Some examples are as follows:

1. The soft Preference Labels $\hat{p}$ is poorly defined. What exactly does the perturbation model refer to? Is it the personalized preference model relative to the population-averaged model? If so, over what is the expectation taken? Why is this $\hat{p}$ readily available data? Should it be $\widehat{p}_{x,y_1,y_2} = 1 - \frac{1}{M}\sum_{i=1}^M \epsilon_i(x,y_1,y_2) \approx 1 - \mathbb{E}[\epsilon(x,y_1,y_2)] = p^\star(y_1>y_2 | x)$ ?

2. The introduction is not easy to parse. Providing a concrete example of the problem being studied would help. For instance, the statement "Nevertheless, many existing RLHF algorithms and Direct Preference Optimization (DPO) [38] variants assume binary deterministic preferences" is confusing, as the usual assumption for DPO is a Bradley-Terry preference model, which is not deterministic. This isn't clarified until the next section.

3. There is no justification for why weighted geometric averaging over the LLM output makes sense. If this approach makes mathematical sense after reparametrization, it would be very helpful to explain this clearly.

4. The explanation in chapter 3.3 is very hard to follow.

5. Are soft preference labels readily available with the current data curation process, or are you advocating for their utilization and thus encouraging the community to propose methods for obtaining them?

**Questions:**

As previously mentioned in the weaknesses section, I am mainly confused about the soft preference labels: are you proposing to obtain these soft preference labels instead of the binary deterministic ones currently used, or are you suggesting that we have been under-exploiting our data, which already contains this information?

Additionally, it would be helpful to explain how this distributional information is philosophically different from the Bradley-Terry model, where preference labels are random draws from a fixed distribution representing population-level preferences.

**Limitations:**

The authors address some limitation in their writing.

---

> ### Author Rebuttal · Authors · 2024-08-07
>
> We thank the reviewer for the constructive feedback.
>
> **> W1 & Q2 (Definition of Soft Preference Labels)**
>
> Thank you for pointing this out. We will update the definition of soft preference labels as follows in the revision:
>
> We assume that the binary preference labels ($l(y_1 \succ y_2 | x) = 1$) are sampled from the Bradley-Terry model preference distribution with the parameter $p^{\star}(y_1 \succ y_2 | x) = \sigma(r^{\star}(x,y_1)-r^{\star}(x,y_2))$. We define soft preference labels as estimates of the true preference probability: $\hat{p}_{x,y_1,y_2}:=\hat{p}(y_1 \succ y_2 | x) \approx p^{\star}(y_1 \succ y_2 | x)$. For instance, we can estimate this via monte-carlo sampling such as:
>
> $\hat{p} = \frac{1}{M}\sum_{i=1}^{M} l_i (l_i \in \{0,1\})$,
>
> which is done via majority voting among $M$ people in practice. The sampled binary preference may sometimes flip with probability $\epsilon$ (i.e. label noise). If the label noise is known, we may take the expectation over the noise such as: $\hat{p} = (1-\frac{1}{M}\sum_{i=1}^{M} \epsilon_i) \frac{1}{M}\sum_{i=1}^{M} l_i+\frac{1}{M}\sum_{i=1}^{M}\epsilon_i \frac{1}{M}\sum_{i=1}^{M} (1-l_i)$, or may ignore the noise if $\epsilon_i$ is small and $M$ is sufficiently large. Alternatively, we can also estimate the soft preference directly via Bradley-Terry models with some reward function. The direct estimation is often adopted in AI feedback with scoring.
>
> **> W2 (Does DPO assume binary deterministic preference?)**
>
> We initially use the term “binary deterministic preferences” because, the objective of DPO and reward models for RLHF is $\max\log p_{\theta}(y_1\succ y_2|x)=\max\log \sigma(r_{\theta}(y_1)-r_{\theta}(y_2))$ (as explained in Section 2, L58&L83), and since $p_{\theta}$ is a probability, the maximization objective forces $p_{\theta}(y_1\succ y_2|x) \rightarrow 1$. In IPO paper (https://arxiv.org/abs/2310.12036), it is pointed out that such a determinism formulation of DPO induces $r_{\theta}(y_1)-r_{\theta}(y_2)\rightarrow\infty$ and causes over-optimization issues. However, as the reviewer pointed out, we agree that the term “deterministic” may cause confusion to the readers. In the revision, we will avoid using “deterministic” in the context, and just mention it as “binary labels” or “binary preference” for better readability.
>
> **> W3 (Clarification of Geometric Averaging)**
>
> Weighted geometric averaging of likelihood is one of the design choices to regularize the objective of DPO variants. For instance, RLHF algorithm based on Nash equilibrium (https://arxiv.org/abs/2312.00886) examined geometric averaging of the current policy and reference policy as a regularization (another was EMA).
>
> Geometric averaging pushes large probability to small when the soft preference is far from 1. Figure 1 (b) on rebuttal PDF provides illustrative examples of geometric-averaged Gaussian distribution. The geometric-averaged winner distribution $\bar{q}_w(x)=q_w(x)^{\hat{p}}q_l(x)^{1-\hat{p}}/Z(x)$ is smoothly regularized when soft preference $\hat{p}$ is small. After the transformation of objective, geometric averaging helps mitigate over-optimization issues in DPO and escape from objective mismatch in cDPO by reducing the scaling factor of gradient from small soft preference samples (as in Section 3.2&3.3).
>
> **> W4 (Clarification of Section 3.3)**
>
> In Section 3.3, we demonstrate an toy example of the objective mismatch issues between text generation and preference modeling, observed in linear interpolation of objective function as done in cDPO. In Figure 1 on the main text, we compare the distribution of learned policy with DPO objectives. Note that we optimize parameterized reward function, and learned policy is analytically recovered. The true reward is a linear function to the index. It shows that cDPO accurately fits the data distribution, which has a larger probability mass on the smaller index (smaller reward), while DPO and GDPO assign a larger probability mass on the larger index (larger reward). The learned distribution of cDPO is desirable from the preference modeling perspective, while it is not desirable from the text generation, because greedy decoding only considers the largest probability mass, which only has a lower reward, during text generation. We also have demonstrated that the same phenomena happen in LLM experiments (Figure 4). We can provide further explanation if needed.
>
> **> W5 & Q1 (Does soft labels help improve the alignment?)**
>
> In the paper, we advocate using soft labels to improve the alignment quality, as they contain more information than binary labels. In addition to AI feedback in this paper, if the existing data has some point-wise scores, soft labels can be easily obtained through direct estimation with Bradley-Terry model. If we can prepare multiple human raters, we can estimate more accurate soft preferences through the majority voting by increasing the pool of rater.
>
> In Table 2 on the main text, we have shown that the dataset with richer soft preference (Plasma Plan) achieves larger performance gain compared to the one with more sparse soft labels (Anthropic Helpful & Harmless). To directly compare the trend, we additionally construct a new preference label dataset on top of Anthropic Helpful and Harmless. In Figure 1 (d) on rebuttal PDF, we show a histogram of soft labels $\hat{p}$ in new preference datasets simulated with AI feedback. We collect competitive paired samples with winner responses of original dataset and from PaLM 2-L to realize richer and more diverse preference distributions that have enough mass around the modest confidence (e.g. $\hat{p}\in[0.7, 0.9)$).
>
> Figure 1 (e) in rebuttal PDF shows the winning rate learned from new preference distribution in Figure 1 (d). The results highlight that rich soft labels help align LLMs better than original dataset with sparse soft labels (esp. notable in Anthropic Harmless; e.g. +12.93% absolute improvement by geometric-averaging on binary winning rate).

---

> ### Author Response · Authors · 2024-08-12
>
> Dear Reviewer,
>
> While this is a last minute before closing discussion period, we would appreciate it if you could check our updates and feel free to raise further questions if you have any. We are happy to address them further. Thank you so much for your time.
>
> Sincerely,
>
> Authors

---

> > ### Comment · Reviewer_GdY5 · 2024-08-13
> >
> > Dear Authors,
> >
> > Thank you for the detailed response. I have read the rebuttals that the authors provided as well as other concurrent discussions, and I am willing to raise my score accordingly since I think the idea is interesting enough.

---

### Official Review · Reviewer_SyyB · 2024-07-15

**Soundness:** 3
**Presentation:** 3
**Contribution:** 3
**Rating:** 7
**Confidence:** 4

**Summary:**

Many existing algorithms for aligning large language models (LLMs) with human preferences assume these preferences are binary and deterministic. However, preferences can vary among individuals and should be represented distributionally to capture subtle relationships between responses. This work introduces distributional soft preference labels and enhances Direct Preference Optimization (DPO) by incorporating a weighted geometric average of the LLM output likelihood in the loss function. This adjustment scales the learning loss based on soft labels, minimizing loss for equally preferred responses. This modification is applicable to any DPO family, addressing the objective mismatch seen in prior work. Experiments using simulated soft preference labels with AI feedback from LLMs show that geometric averaging consistently improves performance on standard alignment benchmarks, particularly with data containing a majority of modestly-confident labels.

**Strengths:**

1. This paper is well written. The notations are clear and the literature review is sufficient.
2. The proposed distributional soft labels combined with weighted geometric averaging of the output likelihood in the loss function can mitigate the mismatch between the text generation and preference modeling.
3. Simulation on the soft preference labels verifies the proposed method in for diversed preference label distributions. This provides insightful evidences and implications for the performance in a practical scenario.

**Weaknesses:**

Given the multiple aspect of human preference, the proposed method is only using a soft label scalar scoring which is established based on the basic assumption of transivity, derived from BT model and its formulation. Therefore, the trade-off between diverse aspects of preferences, especially when the preferences are contradictory, is not explicitly addressed.

**Questions:**

It would be interesting to understand how could a model derived based on transitivity assumption or soft label based scalar ratings be able to reflect multiple aspects of real world preferences, e.g., the balance between helpfulness and harmlessness in Anthropic dataset.

---

> ### Author Rebuttal · Authors · 2024-08-07
>
> We appreciate the reviewer’s thoughtful feedback.
>
> **> W1 & Q1 (Can GDPO handle multiple preference labels at once?)**
>
> Following the reviewer’s suggestion, we finetune the LLM the Anthropic Helpfulness and Harmlessness preference datasets simultaneously to study the balance between different real-world preferences. For example, the Harmlessness dataset instructs the LLM to provide concise refusals (e.g., "I don't know") when content is inappropriate, while the Helpfulness dataset encourages detailed responses, which can be conflicting with each other.
>
> The experimental results are shown in **Figure 1 (a)** in the rebuttal PDF. Soft preference methods appear to outperform vanilla DPO, presumably because of avoiding the over-optimization problem. And we can see that GDPO consistently outperforms all baseline methods, same as our other experiments.
>
> We will add these results in the Appendix in the revision. Please let us know if you have further questions.

---

> > ### Comment · Reviewer_SyyB · 2024-08-12
> >
> > I have read the rebuttal and thank the authors for their detailed and candid responses.
> >
> > It is straightforward to understand that the proposed method outperforms all baseline methods, despite that the structural gap in handling contracting real-world preferences remains unresolved.
> >
> > I maintain my positive opinion on this paper.

---

> ### Author Response · Authors · 2024-08-12
>
> We thank the reviewer for reading the responses and your thoughtful consideration.
>
> As the reviewer pointed out, it is an important direction in practical scenarios to align LLMs to multiple preferences conflicting with each other. In addition to our experimental results, we will include this point in the revision. Thank you again for taking your time.

---

### Author Rebuttal · Authors · 2024-08-07

We sincerely thank the reviewers for the detailed and thoughtful feedback. To address the concerns and questions, we provide a one-page PDF for the figures and tables of additional experiments in addition to our response to each reviewer. Here is a brief overview of the PDF contents:

- **(a)** Finetuning with multiple preferences from both Anthropic Helpful and Harmless datasets (Reviewer **SyyB**)
- **(b)** Illustrative examples of Geometric-averaged Gaussian distribution (Reviewer **GdY5**)
- **(c)** Agreement of human-LLM judge (Reviewer **EtLq**)
- **(d)** Comparison between original and new soft preference distribution to simulate diverse soft labels (Reviewer **GdY5**)
- **(e)** Winning rate with new soft preference distribution presented in (d) (Reviewer **GdY5**)
- **(f)** Log ratio and Estimated reward gap on Plasma Plan and Anthropic Harmlessness among DPO/cDPO/GDPO (Reviewer **msmA & EtLq & 98AM**)
- **(g)** Winning rate under preference noise $\epsilon \in {0.1,0.2,0.3}$ (Reviewer **msmA**)
- **(h)** Online alignment with extra reward model/self-preference (Reviewer **98AM**)
- **(i)** Performance with Gemma-2B, another open-source LLM, instead of PaLM 2-XS (Reviewer **98AM**)

In addition, we provide a theoretical analysis of GDPO in response to Reviewer **msmA**.

Please let us know if there are remaining concerns or questions, which we would be happy to address again.

---

### Decision · Program_Chairs · 2024-09-25

**Decision:**

Accept (poster)

**Comment:**

This paper studies the alignment of LLMs, and proposes to utilize a richer feedback than the usual binary preference labels. The authors assume they have access to distributional soft preference labels, and propose a weighted geometric averaging technique that can be applied to DPO and its variants. The main claim is that their training objective adjusts the scale of learning loss based on soft labels, which can improve performance on standard alignment benchmarks.

The reviewing team all agree that the main idea of the paper is interesting, and that the paper introduces an incremental yet solid contribution to the literature. They have raised several issues—which I will not repeat here—that authors should incorporate in a revised version of the submission. In particular, the authors should provide a clear discussion of why the geometric averaging schemes allow incorporating soft labels.

Furthermore, I recommend the authors discuss where they expect soft preference labels to be derived from in practice. LLM-generated soft labels may not always be reliable (esp. outside of research datasets) and it is unclear what the cost of such finer-grained information would be.